# Al$_{13}^-$ and B@Al$_{12}^-$ superatoms on a molecularly decorated substrate

Masahiro Shibuta[1], Tomoya Inoue[2], Toshiaki Kamoshida[2], Toyoaki Eguchi[3] & Atsushi Nakajima [1,2✉]

Aluminum nanoclusters (Al$_n$ NCs), particularly Al$_{13}^-$ ($n = 13$), exhibit superatomic behavior with interplay between electron shell closure and geometrical packing in an anionic state. To fabricate superatom (SA) assemblies, substrates decorated with organic molecules can facilitate the optimization of cluster–surface interactions, because the molecularly local interactions for SAs govern the electronic properties via molecular complexation. In this study, Al$_n$ NCs are soft-landed on organic substrates pre-deposited with $n$-type fullerene (C$_{60}$) and $p$-type hexa-$tert$-butyl-hexa-$peri$-hexabenzocoronene (HB-HBC, C$_{66}$H$_{66}$), and the electronic states of Al$_n$ are characterized by X-ray photoelectron spectroscopy and chemical oxidative measurements. On the C$_{60}$ substrate, Al$_n$ is fixed to be cationic but highly oxidative; however, on the HB-HBC substrate, they are stably fixed as anionic Al$_n^-$ without any oxidations. The results reveal that the careful selection of organic molecules controls the design of assembled materials containing both Al$_{13}^-$ and boron-doped B@Al$_{12}^-$ SAs through optimizing the cluster–surface interactions.

[1] Keio Institute of Pure and Applied Sciences (KiPAS), Keio University, 3-14-1 Hiyoshi, Kohoku-ku, Yokohama 223-8522, Japan. [2] Department of Chemistry, Faculty of Science and Technology, Keio University, 3-14-1 Hiyoshi, Kohoku-ku, Yokohama 223-8522, Japan. [3] Department of Physics, Graduate School of Science, Tohoku University, 6-3 Aramaki Aza-Aoba, Aoba-ku, Sendai 980-8578, Japan. ✉email: nakajima@chem.keio.ac.jp

Through the deposition of size-selected atomic clusters consisting of a few to thousands of atoms on well-defined substrates, nanostructured surfaces can be produced through bottom-up fabrication, which is a promising method for creating low-dimensional nanomaterials with atomic-scale structural precision[1–4]. The properties of functionalized nanostructured surfaces can be controlled by designing cluster–surface interactions, which facilitates a nanoscale approach to developing nanomaterial-based modified electrodes for application in electrochemistry[5]. The cluster–surface interaction is a fundamental characteristic of such nanostructured materials[3,6], and has been a focus in the preparation of heterogeneous catalysts through control of the physical and chemical properties, size, and dimensionality[7–10]. For example, Haruta indicated the importance of choosing a substrate in enhancing the catalytic activity of gold (Au) nanoparticles for low-temperature CO oxidation[7]. In addition, the substrate acidity has been reported to control the catalytic activity of size-selective platinum (Pt) clusters[10]. In these studies, localized cluster–surface interactions are enhanced using metal oxide substrates[8,9] to avoid the generation of weakly bound nanoclusters (NCs) on a clean surface, because these NCs generally behave as a two-dimensional gas, ultimately resulting in aggregation[6].

Interactions that take place through charge transfer (CT), or more explicitly, electron transfer[11], are important in chemical reactions between two reactant molecules since they lead to the formation of intermolecular CT complexes that exhibit a new electronic transition known as a CT band[12]. Their segregated stacking can lead to molecular electrical conductivity, including superconductivity[13,14]. Such CT processes play an important role in cluster–surface interactions. More specifically, due to the CT interactions with pre-deposited organic molecules on a substrate, the NCs can exist in a monodisperse state on the surface[15,16].

Among various gas phase NCs and their characteristic functionalities explored during the past several decades, NCs formed with a highly symmetrical geometry and an electronically closed shell are known as "superatoms" (SAs), which mimic the chemical properties of atoms with clusters[17–25]. In particular, anionic aluminum (Al) NCs with 13 atoms, i.e., $Al_{13}^-$, are promising candidates for the fabrication of SA assembled nanomaterials[26–31], because $Al_{13}^-$ simultaneously satisfies both icosahedral packing and the electronic shell closing[32,33] of 40 electrons as $(1S)^2(1P)^6(1D)^{10}(2S)^2(1F)^{14}(2P)^6$, thereby facilitating the bottom-up fabrication of nanostructures with desired functionalities, similar to the case of building nanoblocks[34–36].

In this study, we show that the choice of organic substrate can allow molecular control of the CT interactions at the cluster–surface interface and stabilize SAs on the surface. Since the localized interactions between the pre-decorated organic molecules and the deposited NCs are enhanced compared to those of a clean bulk metal or semiconductor substrate, the organic substrate is key to immobilization of the deposited NCs, in which the NC aggregation caused by two-dimensional gas behaviors is suppressed[16,25]. Thus, we deposit $Al_{13}^-$ and boron-doped $B@Al_{12}^-$ SAs[17,26,37,38] on organic substrates of n-type $C_{60}$ and p-type hexa-tert-butyl-hexa-peri-hexabenzocoronene (HB-HBC, $C_{66}H_{66}$ (see Supplementary Fig. 1 and Supplementary Note 1)). Spectroscopic characterization by X-ray photoelectron spectroscopy (XPS) and oxidative reaction measurements of the $Al_{13}^-$ and $B@Al_{12}^-$ SAs on the organic substrates are then conducted to reveal that superatomic behavior can be observed on the p-type organic substrates through CT interactions.

## Results

### Charge state of the $Al_n$ NCs on n-type $C_{60}$ and p-type HB-HBC substrates.

Through magnetron sputtering (MSP) of the Al targets, the generated $Al_n^-$ NCs possessed a mass-to-charge ratio (m/z) predominantly in the range of 200−800 (see Supplementary Fig. 2). With an ion current of 300 pA, samples containing $2.9 \times 10^{13}$ mass-selected NCs (~0.6 monolayers (MLs)) could be prepared within 3 h (see "Methods" section and Supplementary Note 2). The morphology of the deposited NCs on the organic substrate was confirmed by scanning tunneling microscopy (STM) imaging[16,25], wherein the SAs were found to be monodispersively immobilized without aggregation (see Supplementary Fig. 3).

Figure 1a, b show the XPS spectra around Al 2p core levels for (a) $Al_{13}$ on $C_{60}$ and (b) $Al_{13}$ on HB-HBC before (lower) and after (upper) $O_2$ exposure, respectively. The binding energies (BEs) of Al $2p_{3/2}$ for the bulk Al ($Al^0$) and oxidized Al ($Al^{3+}$) have been previously reported (marked by vertical bars in the figure)[39]. As can be seen, without $O_2$ exposure, the Al atoms on the $C_{60}$ substrate are completely oxidized, while the Al atoms on HB-HBC are not oxidized. Following $O_2$ exposure, the Al atoms on $C_{60}$ remain unchanged, while Al atoms on HB-HBC are oxidized to $Al^{3+}$. As shown in Fig. 1c, d, the corresponding O 1s component can be observed in the lower trace of Fig. 1c even without $O_2$ exposure. These results show that the $Al_{13}$ NCs present on the $C_{60}$ substrate are so reactive that the nascent NCs are oxidized after deposition by some residual gas in the vacuum chamber ($<10^{-5}$ Pa) during the deposition process.

The contrasting oxidation behavior of these two systems correlates well with the C 1s XPS peaks from the underlying $C_{60}$ or HB-HBC on highly oriented pyrolytic graphite (HOPG), where the C 1s signals are mainly derived from the topmost molecular layer (Fig. 1e, f). As shown in Fig. 1e, after the deposition of $Al_{13}$ on $C_{60}$, the C 1s peak shifts toward a lower BE by ~0.30 eV. Although $Al_{13}$ is nascently oxidized, the shift to a lower BE shows that an anionic $C_{60}^-$ state is formed by $Al_{13}$ oxides through a CT interaction[16,40]; the degree of shift corresponds well to the formation of $C_{60}^-$ as reported in the literature[41] (see Supplementary Note 3 and Supplementary Fig. 4). A similar C 1s shift has been reported when the alkali-like $Ta@Si_{16}$ SA[25] is deposited on $C_{60}$, wherein a shift attributable to $Ta@Si_{16}^+C_{60}^-$ is observed[40,41], as denoted in Fig. 1e. More quantitatively, when the $C_{60}$-derived C 1s peak is deconvoluted into two peak components corresponding to $C_{60}$ alone (non-interacted) and bound with the $Al_n$ oxide (interacted), the BE of the interacted $C_{60}$ peak is 0.33 eV lower than that of non-interacted $C_{60}$ (Supplementary Fig. 4). In addition to $Al_{13}$, the $Al_7$ NCs deposited on $C_{60}$ is nascently oxidized, as can be observed from the Al 2p XPS spectrum, although $Al_7^+$ is regarded to complete the 2S shell (i.e., 20 e$^-$)[21]. In contrast, after the deposition of 0.6 MLs of $Al_{13}$ on the HB-HBC substrate, the C 1s peak shown in Fig. 1f shifts toward a higher BE by ~0.25 eV. Since a similar behavior can be observed for the deposition of the halogen-like $Lu@Si_{16}$ SA[25] onto HB-HBC, this shift suggests the formation of a cationic $HB-HBC^+$ state, and in turn, an $Al_{13}^-$/HB-$HBC^+$ CT complex.

In addition to $Al_{13}^-$, all $Al_n^-$ NCs ($n = 7-24$) can be size-selectively deposited onto $C_{60}$ and HB-HBC substrates. More specifically, the Al 2p XPS spectra show that these $Al_n$ NCs were successfully deposited onto HB-HBC without undergoing any oxidation reactions (see Supplementary Fig. 5). However, complete oxidation was observed for the $Al_n$ NCs deposited on $C_{60}$. It should be emphasized that this contrast in the reactivity of $Al_n$ results from the different types of organic substrate molecules, i.e., n-type and p-type for $C_{60}$ and HB-HBC, respectively. In the Al 2p XPS spectra for the $Al_n$ NCs on HB-HBC, peaks were observed in the range of 73.0–73.2 eV, which is close to the peak position for bulk Al (i.e., 73.0 eV)[39] (see Supplementary Fig. 6). In addition, the small size dependence is consistent with that in the Al 2p core-level BEs for $Al_n^+$ ($n = 12-15$) obtained from the soft X-ray photoionization efficiency curves[42]. More precisely, the

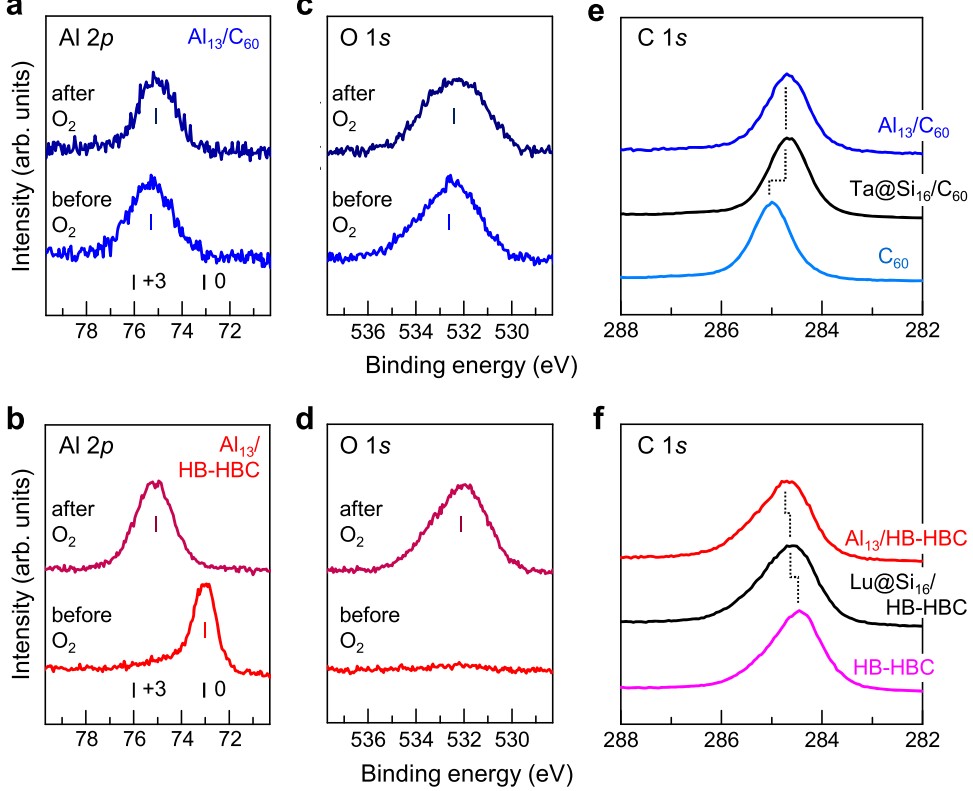

**Fig. 1 XPS spectra for Al₁₃ on the C₆₀ and HB-HBC substrates.** a–d XPS spectra around the Al 2p core levels for **a** Al₁₃ on C₆₀ and **b** Al₁₃ on HB-HBC before (lower) and after (upper) O₂ exposure (at $5 \times 10^{10}$ Langmuir (L = $1.33 \times 10^{-4}$ Pa·s)), along with **c, d** the O 1s spectra for each state. **e, f** The XPS spectra around the C 1s core level for the underlying **e** C₆₀ and **f** HB-HBC are also shown for the deposition of Al₁₃ and Ta@Si₁₆ or Lu@Si₁₆. Reference binding energies (BEs) of Al $2p_{3/2}$ for the bulk Al (Al⁰ and Al³⁺) and O 1s (O²⁻) are marked by vertical bars. The BEs for Al 2p show zerovalent Al⁰ only for Al₁₃ on the HB-HBC substrate before O₂ exposure, while the other BEs are in the vicinity of Al³⁺. After the deposition of 0.6 ML Al₁₃ (blue) or Ta@Si₁₆ (black) on C₆₀, the C 1s peak in (**e**) shifts by ~0.3 eV toward a lower BE from that before deposition (light blue), indicating the presence of an anionic C₆₀⁻ state. After the deposition of 0.6 ML Al₁₃ (red) or Lu@Si₁₆ (black) on HB-HBC, the C 1s peak in (**f**) shifts by ~0.25 eV toward a higher BE from that before deposition (pink), indicating the presence of a cationic HB-HBC⁺ state.

charge states of the Al atoms for the deposited Al$_n$ NCs can be discussed in terms of their Al 2p peak positions; the BEs of Al 2p for all Al$_n$ NCs are slightly higher than that of the bulk Al (zerovalent Al⁰), suggesting that the Al$_n$ NCs on HB-HBC are anionic rather than neutral. Recently, Kambe et al. have reported the Al 2p XPS spectra for several Al$_n$ species ($n = 4, 12, 13, 28,$ and 60) synthesized with dendrimers[43], and they revealed a size-dependent behavior in the Al 2p XPS spectra from 71.2 ($n = 4$) to 72.3 eV ($n = 13$) along with a particular shift of more than 0.6 eV between $n = 12$ and 13. However, our Al 2p spectra exhibit a cluster-size dependence within only 0.3 eV for $n = 7$–24, and no particular peak shift can be observed around $n = 13$. It should be noted here that the peaks in the C 1s XPS spectra for the Al$_n$ NCs on the C₆₀ and HB-HBC/HOPG substrates exhibit a similar contrast shift; namely a decrease in the BE for the Al$_n$ NCs on C₆₀ ($-0.30$ eV) and an increase in the BE for the Al$_n$ NCs on HB-HBC ($+0.25$ eV), with a small size-dependent shift being observed (see Supplementary Fig. 7).

**Oxidative reactivity of Al$_n$ on the HB-HBC substrate.** As shown in Fig. 1b, the Al$_n$ NCs deposited on HB-HBC are oxidized upon O₂ exposure, and the oxidative rates are dependent on the NC size. Figure 2 shows the Al 2p XPS spectra for the Al₁₃ on HB-HBC at several different O₂ exposure amounts (i.e., 0–$5 \times 10^{10}$ L), where the O₂ exposure amounts (in Langmuir units, L = $1.33 \times 10^{-4}$ Pa·s)) are noted on the right-hand side in of the figure. With increasing the amount of O₂ exposure, the intensity of the peak corresponding

to the zerovalent Al⁰ component decreases, while that of the oxidized component Al³⁺ increases along with that of the O 1s component. The oxidative reactivity can therefore be quantitatively evaluated on the basis of its dependence on the O₂ exposure amount from 0 to $1 \times 10^4$ L. It should be noted here that at $1 \times 10^4$ L O₂, the Al⁰ component survives only in the case of $n = 13$ (Supplementary Fig. 5), which is peculiarly unreactive compared to NCs of other sizes and with Al single crystal surfaces, which are completely oxidized when exposed to 400 L O₂ at room temperature[44]. Furthermore, both the Al 2p and O 1s peaks shift to a lower BE when the O₂ exposure amount is increased from $1 \times 10^4$ to $5 \times 10^{10}$ L, thereby implying that a structural change relevant to a phase transition from amorphous to crystalline Al₂O₃ takes place, such as the formation of α- or γ-Al₂O₃[45].

The chemical reactivity of the Al$_n$ NCs toward O₂ gas was then evaluated based on the oxidation rate, $O_{Al_n}$, which is a simple index for investigating the size-dependent behavior of the oxidation reaction. More specifically, the peak area ratio, $R_{Al_n}$, of the non-oxidized component ($S_{Al^0}$) to the oxidized component ($S_{Al^{3+}}$) for the Al 2p spectra is plotted against the logarithm of the O₂ exposure amount in L ($\log_{10} O_2$), and the linear slope is evaluated as $O_{Al_n}$, where $R_{Al_n}$ is expressed as follows:

$$R_{Al_n} = \frac{S_{Al^0}}{S_{Al^0} + S_{Al^{3+}}} \qquad (1)$$

In this analysis, the oxidation of Al atoms by O₂ is modeled in terms of the dissociative adsorption of O₂ on a single crystal Al

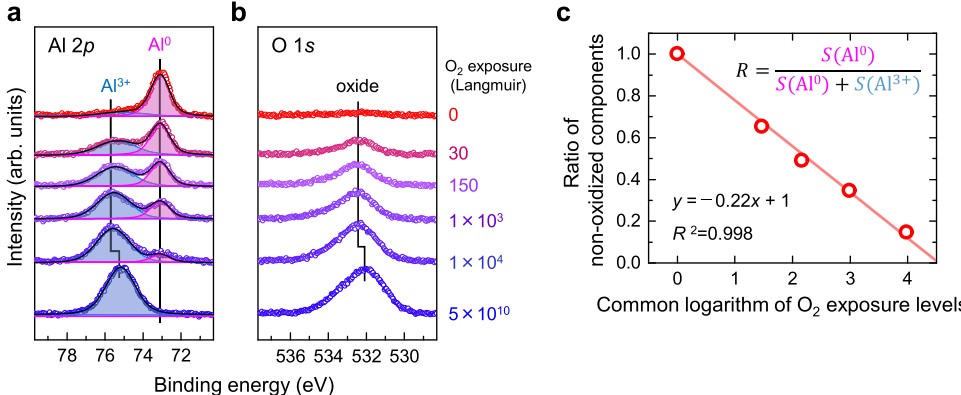

**Fig. 2 Oxidative behaviors in the Al 2p and O 1s XPS spectra for Al$_{13}$ on the HB-HBC substrate. a, b** XPS spectra around **a** Al 2p and (**b**) O 1s. With increasing O$_2$ exposure (from top red to bottom blue), the intensity of the zerovalent component (Al$^0$) decreases, while those of the oxidized component (Al$^{3+}$) and the O 1s component increase accordingly. **c** The oxidative reactivity is evaluated by the slope of the dependence against the logarithmic O$_2$ exposure amount from 0 L to $1 \times 10^4$ L. At the highest exposure of $5 \times 10^{10}$ L, the Al 2p peak shifts to a lower BE, likely due to a structural change relevant to the phase transition of aluminum oxide (see the main text for further details).

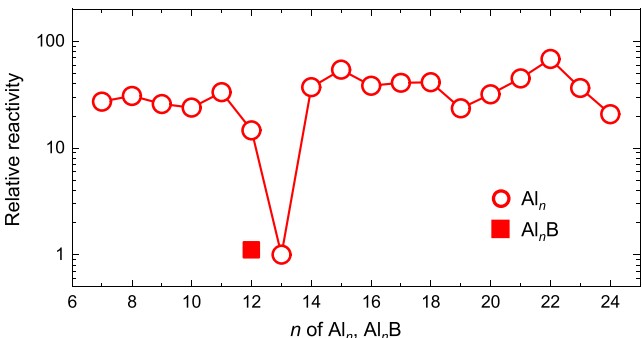

**Fig. 3 Size dependent relative reactivities of Al$_n$ ($n = 7$–24) and B@Al$_{12}$ on the HB-HBC substrate against the O$_2$ exposure.** The oxidative reactivity rates for Al$_n$ on the HB-HBC substrate are plotted (red open circles), and a clear local minimum is observed at $n = 13$ along with a small minimum at $n = 19$, while there is no apparent local minimum at $n = 23$. B@Al$_{12}$ shows a low oxidative reaction rate (red solid square), similar to that of Al$_{13}$.

surface, whose XPS peak appears at a BE close to that of the Al$^{3+}$ component (i.e., $75-76$ eV)[44]:

$$\mathrm{Al}_n \cdot x\mathrm{O}_2 + \mathrm{O}_2 \rightarrow \mathrm{Al}_n \cdot (x+1)\mathrm{O}_2, \ (x = 0, 1, 2, \cdots) \quad (2)$$

The oxidation rates, $O_{\mathrm{Al}n}$, are evaluated by considering the conversion of $1\,\mathrm{L} \rightarrow 1s$ because the exposure amount can be converted to the corresponding reaction time for elementary reactions. The intersection of the linear line with the x-axis in Fig. 2c gives the O$_2$ exposure amount of $V_{\mathrm{Al}13}(\mathrm{O}_2)$ that is required to completely oxidize the Al$_n$ NCs; for Al$_{13}$, $3.45 \times 10^4$ L O$_2$ is obtained as the value of $V_{\mathrm{Al}13}(\mathrm{O}_2)$. The higher the reactivity, the smaller the quantity of oxygen required to completely oxidize the Al$_n$ NCs; for example, $V_{\mathrm{Al}n}(\mathrm{O}_2)$ at $n = 12$ is $2.36 \times 10^3$ L O$_2$, thereby showing that Al$_{12}$ is 14.6 times more reactive than Al$_{13}$ (further details regarding the $O_{\mathrm{Al}n}$ and $V_{\mathrm{Al}n}(\mathrm{O}_2)$ values can be found in Supplementary Note 4 and Supplementary Table 1).

Figure 3 shows the size dependence of the oxidative rates on the Al$_n$ NCs ($n = 7$–24) deposited on the HB-HBC substrate, where the relative reactivity is evaluated by dividing the $V_{\mathrm{Al}n}(\mathrm{O}_2)$ value at $n = 13$ by each individual $V_{\mathrm{Al}n}(\mathrm{O}_2)$ value. A local minimum is clearly found at $n = 13$, and a small local minimum is also found at $n = 19$, while an even–odd alternation relevant to spin conservation[46] observed in the gas phase reaction[17,30,47,48] is not obvious. According to previous experimental and theoretical

works[17,30,38,49–51], electronically stabilized Al$_n$ anions should appear at $n = 19$ and 23 as well as at $n = 13$. Since its interactions with HB-HBC induces an anionic character in the deposited Al$_n$ NCs, Al$_{13}$ and Al$_{19}$ complete their 2P (40 e$^-$) and 1G (58 e$^-$) shells, respectively. However, such stabilization was not observed for Al$_{23}$ despite this species completing its 3S (70 e$^-$) shell (see Fig. 3), and this was attributed to the fact that Al$_{23}$ is geometrically deformed on the substrate owing to its relatively low rigidity having structural $C_s$ symmetry[50–52].

**Oxidative reactivity of B@Al$_{12}$ on the HB-HBC substrate.** The structural rigidity of icosahedral Al$_{13}{}^-$ is demonstrated by the boron (B) doped B@Al$_{12}{}^-$ SAs. Boron belongs to the same group as Al in the periodic table, and it has been reported B@Al$_{12}{}^-$ can be preferentially formed as an SA both experimentally and theoretically because the isoelectronic and geometrically small B atom facilitates relaxation of the icosahedral geometric strain when used as a central atom[37,38,53,54]. Thus, using a B-mixed Al target, B@Al$_{12}{}^-$ was formed by MSP and was deposited onto the C$_{60}$ and HB-HBC substrates (Supplementary Fig. 8).

Figure 4a, b show the XPS spectra around the Al 2p core levels for the B@Al$_{12}$ deposited on C$_{60}$ and the B@Al$_{12}$ deposited on HB-HBC, respectively, before (lower) and after (upper) O$_2$ exposures, similar to the spectra in Fig. 1. These XPS spectra show that Al atoms on C$_{60}$ are substantially oxidized without O$_2$ exposure, but the tailing peak in the Al$^0$ region implies that some Al atoms survive without oxidation. In contrast, the Al atoms of B@Al$_{12}$ deposited on HB-HBC are not oxidized in the same manner as those of Al$_{13}$, as shown in Fig. 1b.

Upon O$_2$ exposure, the Al 2p XPS peak for the Al atoms deposited on C$_{60}$ becomes shaper upon oxidation, while the Al atoms on HB-HBC are sequentially oxidized to Al$^{3+}$. As shown in Fig. 4c, d, the corresponding O 1s component can be observed in the lower trace of Fig. 4c even without O$_2$ exposure, but the peak intensity is lower than that observed for the Al$_{13}$ on C$_{60}$ (see Fig. 1c). In fact, the intensity of the O 1s peak increases with O$_2$ exposure, as shown in Fig. 4c. These results show that the B@Al$_{12}$ NCs on C$_{60}$ are reactive, but that the oxidation rate is surpressed because of the geometrical stabilization induced by B atom encapsulation.

As shown in Fig. 4e, f, the B 1s XPS spectra show the effect of such B atom encapsulation. More specifically, despite a similar oxidative reactivity between the Al and B atoms[55,56], the B 1s peak for the B@Al$_{12}$ on C$_{60}$ shows that a non-oxidized B$^0$

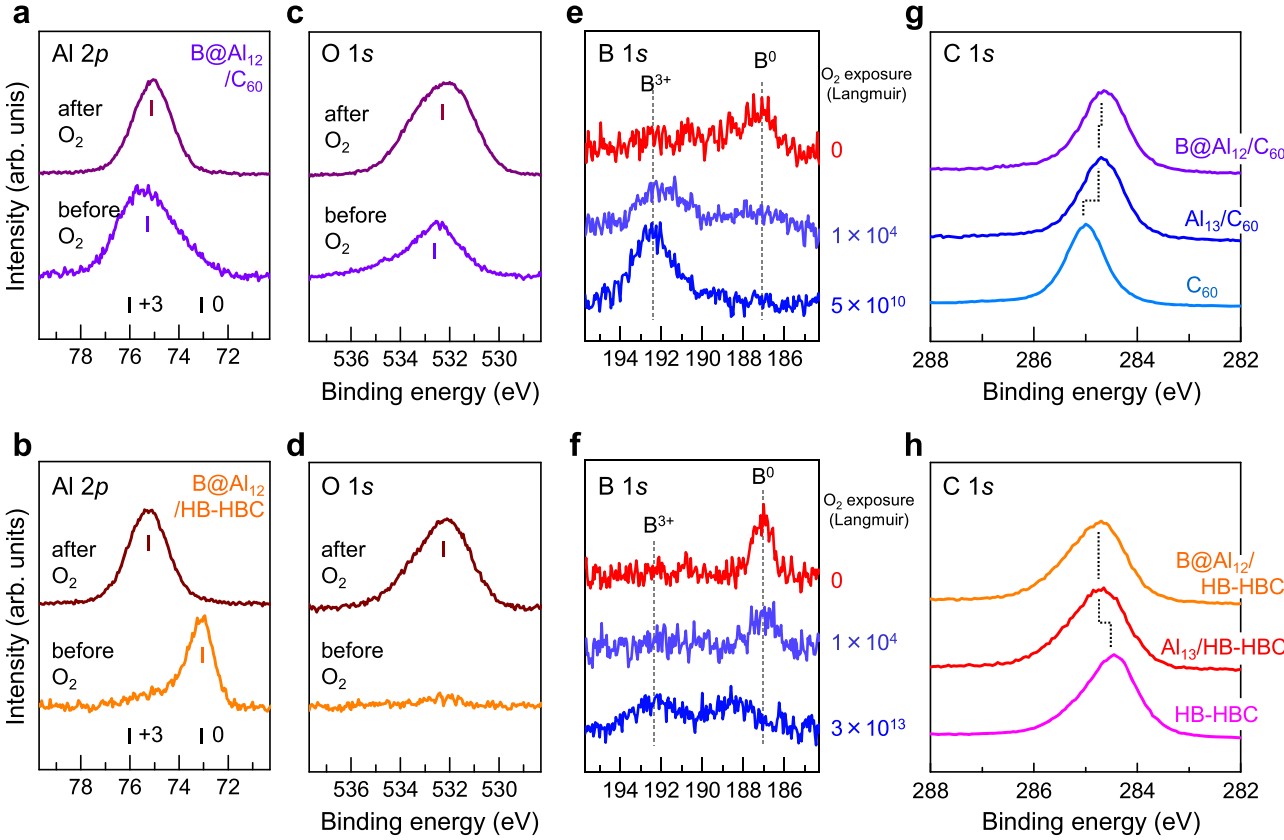

**Fig. 4 XPS spectra for B@Al$_{12}$ on the C$_{60}$ and HB-HBC substrates. a–d** XPS spectra around the core Al 2$p$ levels for **a** B@Al$_{12}$ on C$_{60}$ and **b** B@Al$_{12}$ on HB-HBC before (lower) and after (upper) O$_2$ exposure (at $5 \times 10^{10}$ L), along with **c, d** O 1$s$ for each state. **e, f** XPS spectra around the B 1$s$ core levels for **e** B@Al$_{12}$ on C$_{60}$ **f** B@Al$_{12}$ on HB-HBC before (top) and after (lower) O$_2$ exposure. **g, h** XPS spectra around the C 1$s$ core levels of the underlying **g** C$_{60}$ or **h** HB-HBC are also shown for the depositions of B@Al$_{12}$ and Al$_{13}$. The reference binding energies (BEs) of Al 2$p$ and B 1$s$ for the bulk Al and B (Al$^0$/B$^0$ and Al$^{3+}$/B$^{3+}$) and O 1$s$ (O$^{2-}$) are marked by vertical bars. The BEs for Al 2$p$ show the presence of zerovalent Al$^0$ only for Al$_{13}$ on HB-HBC before O$_2$ exposure, while the other BEs are in the vicinity of Al$^{3+}$, indicating the presence of oxidized Al atoms. After the deposition of 0.6 ML B@Al$_{12}$ (violet) and Al$_{13}$ (blue) on C$_{60}$, the C 1$s$ peak in (**g**) shifts by ~0.3 eV toward a lower BE from that before deposition (light blue), showing an anionic C$_{60}^-$ state. After the deposition of 0.6 ML B@Al$_{12}$ (orange) and Al$_{13}$ (red) on HB-HBC, the C 1$s$ peak in (**h**) shifts by ~0.25 eV toward a higher BE than that before deposition (pink), showing a cationic HB-HBC$^+$ state. Importantly, with B atom doping, B@Al$_{12}$ is stabilized even in the cationic form, as shown by the tailing peak of Al 2$p$ in (**a**) and the non-oxidized B$^0$ component in (**e**).

component can be observed even for the nascent B@Al$_{12}$ on C$_{60}$, showing that the oxidation of B atoms to achieve the B$^{3+}$ state is significantly slower than the corresponding oxidation of Al atoms under O$_2$ exposure. More importantly, the B 1$s$ peak for the B@Al$_{12}$ on HB-HBC can be observed at an O$_2$ exposure amount up to ~$1 \times 10^4$ L, at which point the majority of Al atoms are oxidized. Furthermore, Fig. 4g, h show the C 1$s$ XPS spectra of B@Al$_{12}$ on the C$_{60}$ and HB-HBC substrates, respectively, wherein a behavior similar to that of Al$_{13}$ deposition can be observed. More specifically, for the B@Al$_{12}$ on C$_{60}$, the C 1$s$ peak (Fig. 4g) shifts toward a lower BE by ~0.25 eV, while for the B@Al$_{12}$ on HB-HBC, the C 1$s$ peak (Fig. 4h) shifts toward a higher BE by ~0.25 eV, suggesting the formation of a B@Al$_{12}^-$/HB-HBC$^+$ CT complex.

When the oxidation rate of B@Al$_{12}$ is similarly evaluated based on the peak area ratio of the non-oxidized component ($S_{Al^0}$) to the oxidized component ($S_{Al^{3+}}$) (see Supplementary Fig. 9), the $O_{B@Al12}$ value is the same with the $O_{Al13}$ value within experimental uncertainties, resulting in similar $V_{B@Al12}(O_2)$ and $V_{Al13}(O_2)$ values, as plotted in Fig. 3. Upon B atom encapsulation, all Al atoms become surface Al atoms of the Al$_{12}$ cage, while in contrast, Al$_{13}$ consists of twelve surface Al atoms and one central Al atom. The same oxidative rates observed for B@Al$_{12}$ and Al$_{13}$ therefore indicate that B@Al$_{12}$ is more robust because these

equivalent rates were obtained despite the contribution of the central Al atom of Al$_{13}$.

**Theoretical calculations on the charge distributions for the 13-mer anions and cations.** For Al$_{13}^-$, B@Al$_{12}^-$, Al$_{13}^+$, and B@Al$_{12}^+$, although theoretical calculations have been reported by several groups[36,38,57,58], density functional theory (DFT) calculations are collectively performed to explain the different oxidation behaviors observed for Al$_{13}$/B@Al$_{12}$ on the C$_{60}$ and HB-HBC substrates. The results are presented in Fig. 5, and the Cartesian coordinates are summarized in Supplementary Table 4. For the equilibrium structures, the averaged Al–Al bond lengths are 0.2794 nm for icosahedral Al$_{13}^-$ and 0.2675 nm for icosahedral B@Al$_{12}^-$. The shortened Al–Al bond in B@Al$_{12}^-$ is ascribed to relaxed geometric strains due to the presence of a small central B atom inside the Al$_{12}$ cage. For both Al$_{13}^+$ and B@Al$_{12}^+$, the structural symmetry is lowered, giving $C_1$ symmetry for Al$_{13}^+$ and $C_i$ symmetry for B@Al$_{12}^+$, and this was attributed to the electron deficiency of 2P shell closure.

In terms of the charge distributions of Al$_{13}^-$/B@Al$_{12}^-$ and Al$_{13}^+$/B@Al$_{12}^+$, natural population analysis (NPA) shows that the central Al/B atom is negatively charged, while the surface Al atoms ($\rho$ (Al)) have a positive charge of +0.06/+0.15 for

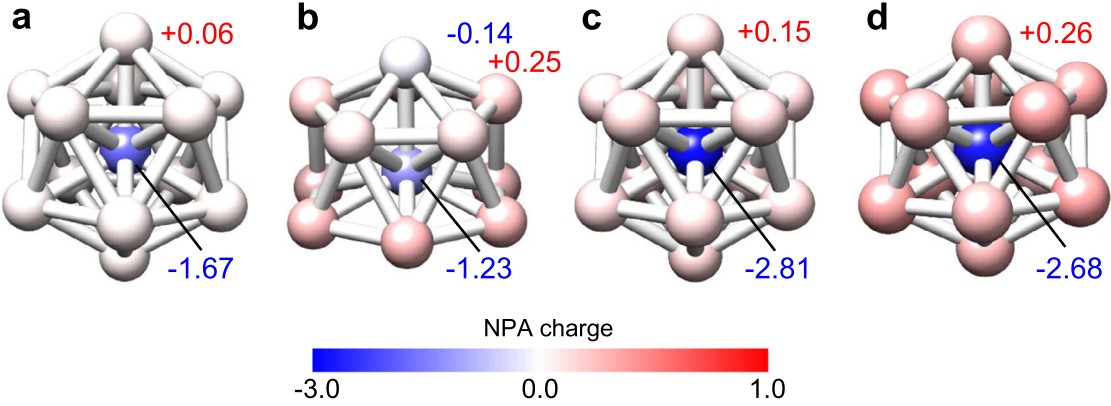

**Fig. 5 Calculated NPA charge distributions for Al₁₃ and B@Al₁₂.** **a–d** Natural population analysis (NPA) distributions for **a** $Al_{13}^-$, **b** $Al_{13}^+$, **c** $B@Al_{12}^-$, and **d** $B@Al_{12}^+$ for the optimized structures using PBE0 with 6-311+G(d) for $Al_{13}^-$ and $B@Al_{12}^-$ or with 6-311G(d) for $Al_{13}^+$ and $B@Al_{12}^+$. Along with the representative values, the charge amount is expressed by the color gradation: positive in red and negative in blue. The icosahedral $I_h$ symmetries for $Al_{13}^-$ and $B@Al_{12}^-$ are lowered to $C_1$ for $Al_{13}^+$ and $C_i$ for $B@Al_{12}^+$ owing to electron deficiency in the 2P shell. In general, a central Al/B atom is negatively charged, while the surface Al atoms are positively charged, with the exception of a top Al atom in the distorted $Al_{13}^+$ in (**b**).

$Al_{13}^-$/$B@Al_{12}^-$. Compared to the central Al atom ($\rho$ (Al) $=$ $-1.67$), the central B atom is more negatively charged ($\rho$ (B) $= -2.68$), and the surrounding twelve Al atoms ($\rho$ (Al) $=$ $+0.15$) are more positively charged than those in $Al_{13}^-$ ($\rho$ (Al) $= +0.06$). For $Al_{13}^+$/$B@Al_{12}^+$, the positive charges are delocalized over all Al atoms in the cluster, with the exception of one negatively charged Al atom. Therefore, these theoretical calculations show common electronic features wherein the negatively charged central atoms are masked by the surrounding Al atoms.

## Discussion

**Charge state of the Al₁₃/B@Al₁₂ deposited on a *p*-type substrate and reaction mechanism.** The charge states of the deposited $Al_{13}$/$B@Al_{12}$ SAs were found to be significantly influenced by the cluster–surface interactions, which in turn are affected by the molecular character of the organic substrate. More specifically, the *p*-type organic substrate of HB-HBC was found to electronically stabilize the halogen-like $Al_{13}$/$B@Al_{12}$ NCs on the surface by donating an electron, which led to electron shell closure.

The ultraviolet photoelectron spectroscopy (UPS) reveals the electronic states of the organic substrates (see the UPS spectrum for HB-HBC in Supplementary Fig. 10). More specifically, before the deposition of $Al_n$, the HOMO energies of the $C_{60}$ and HB-HBC are 2.3 eV[59] and 1.6 eV (at the peak maximum) below the Fermi level ($E_F$), respectively. In addition, the LUMO levels can be accessed by two-photon photoemission spectroscopy; LUMO energies of $C_{60}$ and HB-HBC are 0.7 eV[59] and 1.4 eV above the $E_F$ for $C_{60}$ and HB-HBC, respectively (see Supplementary Fig. 11). These HOMO and LUMO energies indicate that $C_{60}$ (HB-HBC) can be regarded as *n*-type (*p*-type) substrates, wherein $C_{60}$ accepts an electron to its LUMO, while HB-HBC donates an electron from its HOMO. Indeed, the quantitative evaluations carried out for the energetics of CT complexation between $Al_{13}$/$B@Al_{12}$ and $C_{60}$/HB-HBC reasonably explain the formation of $Al_{13}^+C_{60}^-$/ $B@Al_{12}^+C_{60}^-$ and $Al_{13}^-$HB-HBC$^+$/$B@Al_{12}^-$HB-HBC$^+$ (see Supplementary Note 5 and related contents, i.e., Supplementary Fig. 12, Supplementary Table 2, and Supplementary Table 3).

In the context of $O_2$ chemisorption on the surfaces, the adsorbed $O_2$ molecules with two unpaired electrons accept an electron from the surface, forming superoxide ($O_2^-$) or peroxide ($O_2^{2-}$) ions[60,61]. Furthermore, the adsorption energy of two O atoms is larger than the dissociation energy of a single $O_2$

molecule[62], and therefore, O atoms are preferentially bound to the surface via a dissociative electron attachment process[61]. When $O_2$ molecules react with CT complexes on a substrate, the $O_2$ molecules preferentially attack the electron-rich sites of the anions. At the deposition of $Al_{13}^-$/$B@Al_{12}^-$ SAs onto the $C_{60}$ and HB-HBC substrates, as mentioned above, an electron transfer takes place to form $Al_{13}^+C_{60}^-$/$B@Al_{12}^+C_{60}^-$ on $C_{60}$ and $Al_{13}^-$HB-HBC$^+$/$B@Al_{12}^-$HB-HBC$^+$ on HB-HBC. Comparing the electron affinities (*EAs*) of the $C_{60}$ (2.68 eV)[63] and $Al_{13}$/ $B@Al_{12}$ SAs (3.1–3.6 eV)[33,64,65], it is easier to transfer an electron from $C_{60}^-$ to $O_2$ than from $Al_{13}^-$/$B@Al_{12}^-$. In other words, $C_{60}^-$, which is generated by the deposition of $Al_{13}$/$B@Al_{12}$, facilitates the dissociative electron attachment of $O_2$, resulting in the immediate oxidation of $Al_{13}^+$/$B@Al_{12}^+$ cations with $O_2^-$ or $O^-$ through a Coulombic attraction. Therefore, the SA nature of $Al_{13}^-$/$B@Al_{12}^-$ is reinforced by *p*-type molecular decoration, which renders it possible to fabricate assembled surfaces of chemically robust Al-based SAs.

To conclude, we have successfully characterized the series of $Al_n$ NCs deposited on an *n*-type $C_{60}$ and *p*–type HB-HBC substrates. The XPS results reveal that the *n*-type $C_{60}$ substrate possessing a high *EA* withdraws an electron from the $Al_n$ NCs, resulting in a deviation from the electron shell closure. In contrast, the *p*-type HB-HBC substrate donates one electron to the $Al_n$ NCs, generating electronically stable $Al_{13}^-$/$B@Al_{12}^-$ SAs (40 e$^-$). The chemical stabilities of the deposited $Al_n$ examined by step-by-step $O_2$ exposure are shown to be significantly influenced by their charge states on the surface, wherein the stability is enhanced in the 40 e$^-$ systems of $Al_{13}$/HB-HBC and $B@Al_{12}$/HB-HBC along with icosahedral rigidity.

Overall, we have demonstrated the importance of optimizing the cluster–surface interactions to achieve stable depositions of $Al_{13}^-$/$B@Al_{12}^-$ SAs. It has also been demonstrated that the molecular decoration of a substrate aids in controlling the local electronic state through the generation of such cluster–surface interactions. We believe that this molecular strategy for the stable deposition of $Al_{13}$/$B@Al_{12}$ could facilitate the fabrication of SA assemblies for all functional SAs generated in the gas phase.

## Methods

**Sample preparation.** The samples of $Al_n$ or $Al_nB_m$ NCs deposited on organic $C_{60}$ and HB-HBC substrates were prepared in an integrated vacuum chamber, including an MSP source, NC deposition, organic evaporation, and photoelectron spectroscopy systems[16,40,41]. The organic $C_{60}$ and HB-HBC substrates were prepared on cleaned HOPG by thermal evaporation in ultrahigh vacuum (UHV) conditions ($<3 \times 10^{-8}$ Pa). The thicknesses were controlled at 2 and 5 MLs for $C_{60}$

and HB-HBC, respectively, and were monitored using a quartz crystal micro-balance. Commercially available $C_{60}$ (Aldrich, sublimed, 99.9%) was used, while HB-HBC was synthesized (see Supplementary Note 1)[66].

Anionic Al clusters ($Al_n^-$) were generated using an MSP system (Ayabo Corp. nanojima-NAP-01)[25], in which the Al targets were sputtered with $Ar^+$ ions in the MSP aggregation cell. After clustering atomic Al vapors into $Al_n^-$ in a cooled (77 K) He gas flow, the $Al_n^-$ NCs were introduced into a quadrupole (Q) mass filter (Extrel CMS; MAX-16000) through ion optics. The production conditions were optimized by monitoring the mass spectra of $Al_n^-$ (see Supplementary Fig. 2) to maximize the ion intensities at the chosen $m/z$ ratios. The mass-selected $Al_n^-$ NCs were then deposited on the $C_{60}$ and HB-HBC substrates with a mass resolution of $m/\Delta m$ ~70, which was sufficient to exclude the co-deposition of minor products with neighboring $m/z$ values (see Supplementary Fig. 2). The collision energy of the $Al_n^-$ ions was controlled by applying a bias voltage to the substrates (typically +5 V), satisfying the soft-landing conditions (<10 eV/cluster). The number of deposited $Al_n^-$ ions was counted as $2.9 \times 10^{13}$ clusters, where the coverage of $Al_n$ on the substrates was estimated as 0.6 MLs, assuming a deposition area of $2.8 \times 10^{13}$ nm$^2$ (6 mm in diameter) and an $Al_n$ size estimated by a cubic-root interpolation between the sizes of the Al atom ($n = 1$) and the icosahedral $Al_n$ ($n = 13$ and 55) (i.e., 0.62 nm for $n = 7$ and 0.98 nm for $n = 24$ in diameter). The estimated coverage was verified by XPS and UPS measurements with the step-by-step deposition of NCs[40,41]. The deposited samples were transferred to the photoelectron spectroscopy system connected to the cluster deposition system while maintaining UHV conditions. More detailed procedures for sample preparation were described in Supplementary Note 2.

**Photoelectron spectroscopy**. XPS measurements were performed using an Mg Kα ($h\nu = 1253.6$ eV) X-ray source. Photoelectrons emitted from the sample surface were collected with a hemispherical electron energy analyzer (VG SCIENTA, R3000) at a detection angle of 45° from the surface normal. The BE was calibrated using the Au 4 f core level (84.0 eV). It was ensured that no charging effect was observed during any of the XPS measurements. In the XPS analyses, after subtracting the Shirley background, peak fitting was performed by instrumental broadening determined from the Au 4f peak profile (Voigt function with a full width at half maximum (FWHM) of 1.09 eV; the Gaussian and Lorentzian widths were 0.75 and 0.56 eV, respectively). A He–I discharge lamp ($h\nu = 21.22$ eV) was used for the UPS measurements.

To examine the oxidative reactivities of the deposited $Al_n$ NCs, the samples were exposed to $O_2$. The amount of $O_2$ exposure was defined as Langmuir units ($L = 1.33 \times 10^{-4}$ Pa·s). The $O_2$ gas was introduced into the XPS/UPS system using a variable leak valve for low exposure levels ($\leq 10^4$ L). At higher exposure levels (>$10^{10}$ L), the sample was exposed to $O_2$ in a UHV chamber isolated from the XPS/UPS system. All XPS/UPS measurements and $O_2$ exposures were performed at room temperature.

**Density functional theory (DFT) calculations**. Geometry optimizations for the $Al_{13}^-$, $B@Al_{12}^-$, $Al_{13}^+$, and $B@Al_{12}^+$ cluster ions with singlet spin states were performed by DFT implemented in the Gaussian 16 program[67]. All equilibrium geometries were optimized until no imaginary frequencies were found. The hybrid exchange-correlation function PBE0[68,69] was employed at 6-311+G(d) for $Al_{13}^-$ and $B@Al_{12}^-$ and at 6-311G(d) for $Al_{13}^+$ and $B@Al_{12}^+$. Population analyses were performed using NPA[70] for the total electron density obtained at the same level of DFT calculations.

## Data availability

The data that support the findings of this study can be found in the manuscript, Supplementary information, or are available from the corresponding author upon request.

## Code availability

The codes used for the analysis in the current study are available from the corresponding author upon request.

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

## Acknowledgements

We are grateful to Professor Hideyuki Tsukada (Yokohama City University) for supplying the HB-HBC samples, to Professor Takashi Yokoyama (Yokohama City University) for providing information regarding the molecular deposition of HB-HBC, and to Dr. Hironori Tsunoyama for providing some calculation results. This work is partly supported by JSPS KAKENHI of Grants-in-Aid for Scientific Research (A) No. 19H00890 (A.N.) and Scientific Research (C) No. 18K04942 (M.S.), for Challenging Research Nos. 17H06226(A.N.) and 21K18939 (A.N.), and for Transformative Research Areas (A) "Hyper–Ordered Structures Science" (21H05573) (A.N.).

## Author contributions

M.S., T.I., T.K., T. E., and A.N. contributed to the experimental processes. M.S., T.I., and A.N. carried out the simulations and theoretical interpretations. A.N. supervised the overall project. All authors have given approval to the final version of the manuscript.

## Competing interests

A.N. is an inventor on JAPAN patent JP 5493139, submitted by the JST agency and Ayabo Corp., which covers a nanocluster generator. The remaining authors declare no competing interests.
