## [Peer Review File · Nature Communications]

Al13– and B@Al12– superatoms on a molecularly decorated substrateREVIEWER COMMENTS

Reviewer #1 (Remarks to the Author):

Dr. Nakajima reports a XPS study of Al₁₃⁻ and B@Al₁₂⁻ clusters deposited on a molecularly decorated substrate, as well as their reactions with oxygen. They discussed the charge state and O₂ chemisorption mechanism. This manuscript could be publishable but major revisions needed. The deficiency and a few comments are given below.

A proper citation to “superatoms” needs to be added. (J. Phys. Chem. C 2009, 113 (7), 2664-2675.; J. Phys. Chem. Lett. 2011, 2 (9), 1062-1069...). The Ref.21 could be wrong; the right one about superatom by this author may be: Acc. Chem. Res. 2014, 47 (10), 2931-2940.

Also, the literature about 13-atom cluster and the doped clusters is not properly referred to. Some colleagues' work, in my opinion, could be pertinent to this study and may be helpful for the related discussion. The literatures I am referring to can be found at Coord. Chem. Rev. 2019, 400, 213053; and J. Chem. Phys. 2008, 128 (2), 024305; Coord. Chem. Rev. 2020, 403, 213095.

It is a lack of characterization on the cluster deposition. Is it monodispersed or aggregated on the different substrates? What is the essential difference of the two substrates (C₆₀ and HB-HBC)?

In Supplementary Table 2, it is suggested to provide the calculated values at the PBE0 level of theory (chosen in this study) to evaluate the reliability of calculation results. Whether or not the dispersion correction was included to describe the noncovalent interactions?

The authors demonstrated “molecular decorations of a substrate aid in controlling the local electronic state through the cluster–surface interaction”, while the “cluster–surface interactions” were not properly discussed in the context. It could be helpful to discuss how the cluster and surface interact and how electron transfer between them? An interaction diagram or charge decomposition analysis may be helpful.

A few other literatures about the ‘superatom assemble’ and “soft-landing” can be referred to, such as ACS Nano 2009, 3 (2), 244-255; Coord. Chem. Rev. 2021, 429, 213643; Mass Spectrom. Rev. 2016, 35 (3), 439-79.

Reviewer #2 (Remarks to the Author):

People have been proposing for years that closed shell clusters like Al₁₃⁻ should behave like superatoms, and behavior like low reactivity and closed shell electronic spectra in the gas phase have been demonstrated for years. Demonstrating that the clusters can retain “magic” closed shell behavior in the condensed phase is a much more difficult problem because aluminum clusters are quite reactive and also tend to agglomerate unless they are kept well isolated. This paper presents clear evidence that Al₁₃⁻ is stable at reasonably high coverages (i.e., at least somewhat agglomeration resistant) when they are deposited on appropriate p-type organic supporting layers (HB-HBC), but are highly reactive toward oxidation even from background reactions, when deposited on C₆₀ layers. Indeed, all size Al_n⁻ clusters deposited on the HB-HBC layer are stable with respect to oxidation by small exposures of background gases, demonstrating the strong influence of the support on the cluster chemistry. For large deliberate O₂ exposures, however, the Al₁₃⁻/HB-HBC sample was clearly much less reactive than any of the other Al_n⁻ sizes in the n=7-24 range, i.e., there is something magically stable about Al₁₃⁻, when deposited on the appropriate organic supporting layer. Similar behavior is shown for the B@Al₁₂⁻ anion, which should also have the closed electronic shell, but obviously involves a chemically rather different B heteroatom. On C₆₀, the Al component oxidizes even from background gas, but B atom only oxidizes for large O₂ exposures. On HB-HBC, both the Al

and B atoms remain unoxidized with just exposure to background gas, but oxidize for large O₂ exposures.

DFT calculations and UPS spectra were used to probe the charge distribution and how it changes on different supports. This supports the conclusion that electron transfer to or from the clusters on the two supports is important in stabilizing the anion, and promoting oxidation by subsequent O₂ exposure.

Overall, the experiments show that the cluster-support interaction is critical in stabilizing Al clusters as superatoms, and that superatom behavior can, indeed, be maintained for condensed phase clusters on appropriate supports at moderately high coverages. (Although I imagine that aggregation is still a problem at higher loadings). I think that the “electronic properties of clusters” and “supported cluster/cluster-surface interactions” communities will find this paper quite interesting and I support publishing it with the following caveat.

There are many English problems, most of which should be easily corrected by the Nature editorial staff, but some of which might need to be corrected by the authors working together with a native English speaking editor to avoid introducing errors. For example, on page 3 the manuscript states: “The results show that Al₁₃ NCs on C₆₀ are too reactive to be nascently oxidized with some residual gas in a vacuum chamber”. This wording is confusing because “too reactive” suggests that something didn’t happen, whereas the authors mean to say that it did. This could be re-written as “The results show that Al₁₃ NCs on C₆₀ are so reactive that the nascent clusters are oxidized after deposition by some residual gas in the vacuum chamber”

Reviewer #3 (Remarks to the Author):

The authors have studied the soft-landing of Al_n (n=7-14) clusters and B@Al₁₂ on n-type C₆₀ and p-type C₆₆H₆₆ and characterized the electronic states of Al_n with X-ray photoelectron spectroscopy and chemically oxidative measurements. Density functional theory-based calculations are performed to study the charge distribution in Al₁₃⁻, Al₁₃⁺, B@Al₁₂⁻, and B@Al₁₂⁺ clusters. The charge state and the oxidative behavior of the clusters are found to depend on the organic substrate. Studies of clusters supported on different kinds of substrates and an understanding of their cluster-surface interaction is an important field, particularly with respect to forming cluster-assembled materials. The authors are experts in this field and the paper is well written. While these results deserve to be published, I do not believe Nature Communications is the right journal. First, reactivity of Al_n clusters and in particular the chemical inertness of Al₁₃⁻ (Fig. 3) is well-known. Similarly, calculations of neutral and charged Al_n and B@Al₁₂ clusters (Fig. 5) have been carried out before. What is new here is that the interaction and charge-transfer are shown to depend upon the substrate. As such, this is expected. I recommend a more specialized journal. It will be useful to present the ionization potential of C₆₆H₆₆ and to explain why the charge distribution takes the form Al₁₃⁺+C₆₀⁻ since the electron affinity of Al₁₃ is larger than that of C₆₀.

Reviewer #1

Dear Reviewer

First of all, we would like to thank the reviewer for providing a positive evaluation of our manuscript in addition to fruitful suggestions to improve our article. We have revised the original manuscript in accordance with the reviewer's comments, and the revised sections are provided in red type in the revised manuscript. Detailed point-by-point responses to the reviewer's comments (**in bold**) can be found below.

Sincerely yours,

Atsushi Nakajima

Comments:

Dr. Nakajima reports a XPS study of Al_{13}^- and $B@Al_{12}^-$ clusters deposited on a molecularly decorated substrate, as well as their reactions with oxygen. They discussed the charge state and O_2 chemisorption mechanism. This manuscript could be publishable but major revisions needed. The deficiency and a few comments are given below.

A proper citation to "superatoms" needs to be added. (J. Phys. Chem. C 2009, 113 (7), 2664-2675.; J. Phys. Chem. Lett. 2011, 2 (9), 1062-1069...). The Ref.21 could be wrong; the right one about superatom by this author may be: Acc. Chem. Res. 2014, 47 (10), 2931-2940.

Also, the literature about 13-atom cluster and the doped clusters is not properly referred to. Some colleagues' work, in my opinion, could be pertinent to this study and may be helpful for the related discussion. The literatures I am referring to can be found at Coord. Chem. Rev. 2019, 400, 213053; and J. Chem. Phys. 2008, 128 (2), 024305; Coord. Chem. Rev. 2020, 403, 213095.

(Our reply)

Thank you for your suggestions for important references that should be included in our revised manuscript. As suggested by the reviewer, the references relevant to "superatoms" and "13-atom clusters" have been added/replaced in the revised manuscript. Among three references that the reviewer kindly suggested, only two of these, namely *Coord. Chem. Rev.* **2019**, 400, 213053; and *J. Chem. Phys.* **2008**, 128 (2), 024305, have been added, since the total number of references cannot exceed 70 as per the guidelines of *Nature Communications*.

Added references

"Superatoms" references

[as Ref. 17] Castleman Jr., A. W. & Khanna, S. N. Clusters, superatoms, and building blocks of new materials. *J. Phys. Chem. C* **113**, 2664–2675 (2009).

[As Ref. 18] Castleman Jr., A. W. From elements to clusters: The periodic table revisited. *J. Phys. Chem. Lett.* **2**, 1062–1069 (2011).

[As Ref. 21 (Replaced from Ref. 21 in original manuscript (*Chem. Rev.* **116**, 14456–14492 (2016)))]
Luo, Z. & Castleman Jr., A. W. Special and general superatoms. *Acc. Chem. Res.* **47**, 2931–2940 (2014).

"13-Atom cluster" references

[As Ref. 29] Pal, R., Cui, L.-F. Bulusu, S., Zhai, H.-J., Wang, L.-S. & Zeng, X. C. Probing the electronic and structural properties of doped aluminum clusters: $M Al_{12}^-$ ($M = Li, Cu, \text{ and } Au$). *J. Chem. Phys.* **128**, 024305 (2008).

[As Ref. 31] Yin, B. & Luo, Z. Thirteen-atom metal clusters for genetic materials. *Coord. Chem. Rev.* **400**, 213053 (2019).

It is a lack of characterization on the cluster deposition. Is it monodispersed or aggregated on the different substrates? What is the essential difference of the two substrates (C₆₀ and HB-HBC)?

(Our reply)

Thank you for these important questions. We note that the target Al_n nanoclusters, including the Al_{13} superatom species, are immobilized on both organic substrates, i.e., HB-HBC and C_{60} . We consider that pre-decoration with organic molecules of the substrates (organic substrate) is key to the successful deposition and immobilization of the nanoclusters, since the localized interactions between the pre-decorated organic molecules and the deposited nanoclusters are enhanced compared to that of a clean bulk metal or semiconductor substrate, where the freedom for so-called two-dimensional gas behavior is suppressed. In addition, in our previous study, scanning tunneling microscopy (STM) imaging revealed that metal-encapsulating silicon caged superatoms ($M@Si_{16}$) are immobilized on C_{60} and HB-HBC substrates, but that these species undergo aggregation on a graphite (HOPG) substrate (see below figures reproduced from Refs. 16 and 40, as well as Supplementary Figure 3 shown below). Similarly, silver nanoclusters (Ag_n) can be immobilized on a C_{60} substrate, as reported in *Adv. Funct. Mater.*, **24**, 1202 (2014)..

Figure (left) Ta@Si₁₆/C₆₀/HOPG and (right) Ta@Si₁₆/naked HOPG

(Left from Fig. 1(a) in Ref. 16) An STM image (50 × 30 nm²) of C₆₀/HOPG surface obtained after depositing Ta@Si₁₆ cations, showing monodisperse immobilization of Ta@Si₁₆ NCs onto C₆₀-terminated surfaces; bright dots of Ta@Si₁₆ with height of ~0.8 nm can be observed on dark lattice patterns of C₆₀.

(Right from Fig. 2 in Ref. 40). (a) An STM image (120 × 120 nm²) of Ta@Si₁₆ deposited a naked HOPG surface, showing Ta@Si₁₆ aggregation (Height profile taken along a blue-dotted line is given in Ref. 40).

In accordance with the reviewer comments, the importance of the pre-decoration of a substrate with organic molecules has been emphasized in the revised manuscript. In addition, we have added the STM images for Lu@Si₁₆/HB-HBC in the Supplementary Information (Supplementary Fig. 3), since no STM images for nanocluster immobilization on an HB-HBC substrate have been previously published. It should be noted that Lu@Si₁₆ (67 e⁻) is also a halogen-like superatom as well as Al₁₃ due to the lack of the single electron required to complete its electron shell. However, we note that it is not possible to obtain an STM image for Al₁₃⁻ deposition due to the disassembly of the STM instrument. However, we can conclude that the HB-HBC substrate immobilizes Al₁₃ monodispersively in a similar manner to Lu@Si₁₆.

Original manuscript (Page 3, Lines 60-61)

“In this study, we show that the choice of organic substrate can molecularly control CT interactions at the cluster–surface and immobilize Al₁₃⁻ SAs^{17,26}.”

Revised manuscript (Page 3, Lines 60–65)

“In this study, we show that the choice of organic substrate can allow molecular control of the CT interactions at the cluster–surface ~~and immobilize Al₁₃⁻ SAs^{17,26}~~–interface and stabilize SAs on the surface. Since the localized interactions between the pre-decorated organic molecules and the deposited NCs are enhanced compared to those of a clean bulk metal or semiconductor substrate, the organic substrate is key to immobilization of the deposited NCs, in which the NC aggregation caused

by two-dimensional gas behaviors is suppressed^{16,25}.”

Added (Page 3, Lines 77–79)

“The morphology of the deposited NCs on the organic substrate was confirmed by scanning tunneling microscopy (STM) imaging^{16,25}, wherein the SAs were found to be monodispersively immobilized without aggregation (see Supplementary Fig. 3).”

Added (Supplementary Information)

“Supplementary Fig. 3. STM images of Lu@Si₁₆ superatoms deposited on the HB-HBC substrate. (a) wide (200 × 200 nm²) and (b) molecular scale (50 × 50 nm²) STM images of Lu@Si₁₆ (0.3 MLs) on the HB-HBC (5 MLs)/HOPG substrate (3 mm off-centered from the nanocluster deposition beam with 6 mm diameter). Well-aligned protrusions due to the HB-HBC substrate with hexagonal lattice patterns ($a = b = 1.79$ nm) are observed in (b), where the lattice constant is consistent with the in-plane size of the HB-HBC molecule (Supplementary Fig. 1). This means that the HB-HBC film is grown in a flat conformation on the HOPG substrate. Bright dots correspond to the deposited Lu@Si₁₆, showing the successful immobilization of nanoclusters in a monodisperse manner. Note that Lu@Si₁₆ (67 e⁻ valence electrons) is a halogen-like superatom,^{1,2} as is Al₁₃, owing to the lack of a single electron to close the electron shell. The imaging conditions of the tip voltage and the tunneling current are -2.0 V and 0.2 pA for (a), and -2.0 V and 0.5 pA for (b), respectively.”

Added References (Refs. 1 and 2 in the Supplementary Information)

1. Koyasu, K., Atobe, J., Akutsu, M., Mitsui, M. & Nakajima, A. Electronic and geometric stabilities of clusters with transition metal encapsulated by silicon. *J. Phys. Chem. A* **111**, 42–49 (2007).
2. Koyasu, K., Atobe, J., Furuse, S. & Nakajima, A. Anion photoelectron spectroscopy of transition metal- and lanthanide metal-silicon clusters; MSi_n⁻ ($n = 6-20$). *J. Chem. Phys.* **129**, 214301 (2008).

In Supplementary Table 2, it is suggested to provide the calculated values at the PBE0 level of theory (chosen in this study) to evaluate the reliability of calculation results. Whether or not the dispersion correction was included to describe the noncovalent interactions?

(Our reply)

Thank you for this comment and suggestion. For the calculations carried out in relation to the Al_{13} and B@Al_{12} clusters, we note that the dispersion correction does not affect the calculated ionization energy and electron affinity values. When we calculated these values using the dispersion correction (PBE0/6311+G(d)) and without the dispersion correction (PBE0/6311G(d)), the differences between values did not exceed 0.01 eV. Although the dispersion correction is generally included for calculations based on anions, we found that the effect was particularly small, i.e., within 0.01 eV, for our aluminum nanoclusters. More specifically, for Al_{13} , the adiabatic and vertical ionization energies were calculated to be 5.91 (5.90) eV and 6.92 (6.91) eV with PBE0/6-311 (G(d) PBE0/6-311+G(d)), in which PBE0/6-311+G(d) includes the dispersion effect. We have therefore not revised any parts of the manuscript with relation to this point.

The authors demonstrated "molecular decorations of a substrate aid in controlling the local electronic state through the cluster-surface interaction", while the "cluster-surface interactions" were not properly discussed in the context. It could be helpful to discuss how the cluster and surface interact and how electron transfer between them? An interaction diagram or charge decomposition analysis may be helpful.

(Our reply)

Thank you for your kind suggestion. We believe that a local charge transfer (CT) interaction (complexation) between the nanoclusters and the organic molecules on the substrate surfaces enables optimization of the charge state of the deposited species, where the Al_{13} (and B@Al_{12}) superatom could be chemically stabilized on the HB-HBC substrate to form $\text{Al}_{13}^{-}\text{HB-HBC}^{+}$. Although no detailed description has been provided in the main text, the CT interactions are discussed in the Supplementary Note 3. In addition, as shown in Supplementary Fig. 12, the potential curves of the complexation between the $\text{Al}_{13}/\text{B@Al}_{12}$ superatoms and the organic molecules of $\text{C}_{60}/\text{HB-HBC}$ are shown, starting from their neutral species. Based on the accumulated data for the ionization energy and the electron affinity for the Al_{13} , B@Al_{12} , C_{60} , and HB-HBC components, as tabulated in the supplementary information, the endothermic dissociation limits of the corresponding cations and anions were evaluated. The energetics clearly account for the favorable combination of superatom–molecule complexes through charge transfer. However, in the original manuscript, the description regarding superatom–molecule complexation

had been mentioned only in Supplementary Note 3, and then the Supplementary Fig. 12, Table 2, and Table 3 have now been explicitly mentioned in the revised manuscript.

Original manuscript (Page 10, Line 254)

“see Supplementary Note 3”

Revised manuscript (Page 10, Lines 254–255)

“...(see Supplementary Note 3 **and related contents, i.e., Supplementary Fig. 12, Table 2, and Table 3).**”

A few other literatures about the "superatom assemble" and "soft-landing" can be referred to, such as ACS Nano 2009, 3 (2), 244-255; Coord. Chem. Rev. 2021, 429, 213643; Mass Spectrom. Rev. 2016, 35 (3), 439-79.

(Our reply)

Thank you for your kind suggestions. These references have now been added accordingly:

Added references

References for the "superatom assembly"

[As Ref. 35]

Claridge, S. A., Castleman Jr., A. W., Khanna, S. N., Murray, C. B., Sen, A. & Weiss, P. S. Cluster-assembled materials. *ACS Nano* **3**, 244–255 (2009).

[As Ref. 36]

Yin, B. & Luo, Z. Coinage metal clusters: From superatom chemistry to genetic materials. *Coord. Chem. Rev.* **429**, 213643 (2021).

References for the "soft-landing" technique

[As Ref. 4]

Johnson, G. E., Gunaratne, D. & Laskin, J. Soft- and reactive landing of ions onto surfaces: concepts and applications. *Mass Spectrom. Rev.* **35**, 439–79 (2016).

We hope that these responses and revisions are satisfactory, and we would like to thank you once again for your valuable comments and suggestions.

Sincerely yours,
Atsushi Nakajima

Reviewer #2

Dear reviewer

First of all, we would like to thank the reviewer for providing a positive evaluation of our manuscript in addition to fruitful suggestions to improve our article. We have revised the original manuscript in accordance with the reviewer's comments, and the revised sections are provided in red type in the revised manuscript. Detailed point-by-point responses to the reviewer's comments (**in bold**) can be found below. With regards to the issues related to English language usage, our manuscript has been carefully revised by a native English speaker, and so we hope that the reviewer will now find our article exhibits an improved readability and text flow from a language point of view.

Sincerely yours,
Atsushi Nakajima

People have been proposing for years that closed shell clusters like Al_{13}^- should behave like superatoms, and behavior like low reactivity and closed shell electronic spectra in the gas phase have been demonstrated for years. Demonstrating that the clusters can retain "magic" closed shell behavior in the condensed phase is a much more difficult problem because aluminum clusters are quite reactive and also tend to agglomerate unless they are kept well isolated. This paper presents clear evidence that Al_{13}^- is stable at reasonably high coverages (i.e., at least somewhat agglomeration resistant) when they are deposited on appropriate p-type organic supporting layers (HB-HBC), but are highly reactive toward oxidation even from background reactions, when deposited on C_{60} layers.

Indeed, all size Al_n^- clusters deposited on the HB-HBC layer are stable with respect to oxidation by small exposures of background gases, demonstrating the strong influence of the support on the cluster chemistry. For large deliberate O_2 exposures, however, the $Al_{13}^-/HB-HBC$ sample was clearly much less reactive than any of the other Al_n^- sizes in the $n = 7-24$ range, i.e., there is something magically stable about Al_{13}^- , when deposited on the appropriate organic supporting layer.

Similar behavior is shown for the $B@Al_{12}^-$ anion, which should also have the closed electronic shell, but obviously involves a chemically rather different B heteroatom. On C_{60} , the Al component oxidizes even from background gas, but B atom only oxidizes for large O_2 exposures. On HB-HBC, both the Al and B atoms remain unoxidized with just exposure to background gas, but oxidize for large O_2 exposures.

DFT calculations and UPS spectra were used to probe the charge distribution and how it changes

on different supports. This supports the conclusion that electron transfer to or from the clusters on the two supports is important in stabilizing the anion, and promoting oxidation by subsequent O₂ exposure.

Overall, the experiments show that the cluster-support interaction is critical in stabilizing Al clusters as superatoms, and that superatom behavior can, indeed, be maintained for condensed phase clusters on appropriate supports at moderately high coverages. (Although I imagine that aggregation is still a problem at higher loadings). I think that the "electronic properties of clusters" and "supported cluster/cluster-surface interactions" communities will find this paper quite interesting and I support publishing it with the following caveat.

(Our reply)

Thank you for your positive evaluation. With regards the NC aggregation, the importance of substrate pre-decoration with organic molecules has been emphasized in the revised manuscript. In addition, we have added the STM images for Lu@Si₁₆/HB-HBC in the Supplementary Information (Supplementary Fig. 3), since no STM images for nanocluster immobilization on an HB-HBC substrate have been previously published. It should be noted that Lu@Si₁₆ (67 e⁻) is also a halogen-like superatom due to the lack of the single electron required to complete its electron shell. However, we note that it is not possible to obtain an STM image for Al₁₃⁻ deposition due to the disassembly of the STM instrument. However, we can conclude that the HB-HBC substrate immobilizes Al₁₃ monodispersively in a similar manner to Lu@Si₁₆.

Added (Page 3, Lines 77–79)

“The morphology of the deposited NCs on the organic substrate was confirmed by scanning tunneling microscopy (STM) imaging^{16,25}, wherein the SAs were found to be monodispersively immobilized without aggregation (see Supplementary Fig. 3).”

Added (Supplementary Information)

“Supplementary Fig. 3. STM images of Lu@Si₁₆ superatoms deposited on the HB-HBC substrate. (a) wide (200 × 200 nm²) and (b) molecular scale (50 × 50 nm²) STM images of Lu@Si₁₆ (0.3 MLs) on the HB-HBC (5 MLs)/HOPG substrate (3 mm off-centered from the nanocluster deposition beam with 6 mm diameter). Well-aligned protrusions due to the HB-HBC substrate with hexagonal lattice patterns ($a = b = 1.79$ nm) are observed in (b), where the lattice constant is consistent with the in-plane size of the HB-HBC molecule (Supplementary Fig. 1). This means that the HB-HBC film is grown in a flat conformation on the HOPG substrate. Bright dots correspond to the deposited Lu@Si₁₆, showing the successful immobilization of nanoclusters in a monodisperse manner. Note that Lu@Si₁₆ (67 e⁻ valence electrons) is a halogen-like superatom,^{1,2} as is Al₁₃, owing to the lack of a single electron to close the electron shell. The imaging conditions of the tip voltage and the tunneling current are -2.0 V and 0.2 pA for (a), and -2.0 V and 0.5 pA for (b), respectively.”

Added References (Refs. 1 and 2 in the Supplementary Information)

1. Koyasu, K., Atobe, J., Akutsu, M., Mitsui, M. & Nakajima, A. Electronic and geometric stabilities of clusters with transition metal encapsulated by silicon. *J. Phys. Chem. A* **111**, 42–49 (2007).
2. Koyasu, K., Atobe, J., Furuse, S. & Nakajima, A. Anion photoelectron spectroscopy of transition metal- and lanthanide metal-silicon clusters; MSi_n⁻ ($n = 6-20$). *J. Chem. Phys.* **129**, 214301 (2008).

There are many English problems, most of which should be easily corrected by the Nature editorial staff, but some of which might need to be corrected by the authors working together with a native English speaking editor to avoid introducing errors. For example, on page 3 the manuscript states: "The results show that Al₁₃ NCs on C₆₀ are too reactive to be nascently oxidized with some residual gas in a vacuum chamber". This wording is confusing because "too reactive" suggests that something didn't happen, whereas the authors mean to say that it did. This could be re-written as "The results show that Al₁₃ NCs on C₆₀ are so reactive that the nascent clusters are oxidized after deposition by some residual gas in the vacuum chamber"

(Our reply)

Thank you for pointing out this error. We have now revised the indicated sentence, and the full manuscript has been polished by an editing service prior to its resubmission.

Original manuscript (Page 4, Lines 79-80)

“The results show that Al₁₃ NCs on C₆₀ are too reactive to be nascently oxidized with some residual gas in a vacuum chamber (<10⁻⁵ Pa) during the deposition.”

Revised manuscript (Page 4, lines 87–90)

“These results show that the Al₁₃ NCs present on the C₆₀ substrate are so reactive that the nascent clusters are oxidized after deposition by some residual gas in the vacuum chamber ~~too reactive to be nascently oxidized with some residual gas in a vacuum chamber~~ (<10⁻⁵ Pa) during the deposition process.”

We hope that these responses and revisions are satisfactory, and we would like to thank you once again for your valuable comments and suggestions.

Sincerely yours,
Atsushi Nakajima

Reviewer #3

Dear Reviewer

First of all, we would like to thank the reviewer for reading and evaluating our manuscript. However, despite the reviewer's comments relating to our article, we still believe that the present research is suitable for publication in a world-renowned journal such as *Nature Communications*, as justified below. Furthermore, we have revised the manuscript in accordance with the reviewer's comments, and the revised sections are provided in red type in the revised manuscript. Detailed point-by-point responses to the reviewer's comments (**in bold**) can be found below.

The authors have studied the soft-landing of Al_n ($n=7-14$) clusters and $B@Al_{12}$ on n-type C_{60} and p-type $C_{66}H_{66}$ and characterized the electronic states of Al_n with X-ray photoelectron spectroscopy and chemically oxidative measurements. Density functional theory-based calculations are performed to study the charge distribution in Al_{13}^- , Al_{13}^+ , $B@Al_{12}^-$, and $B@Al_{12}^+$ clusters. The charge state and the oxidative behavior of the clusters are found to depend on the organic substrate. Studies of clusters supported on different kinds of substrates and an understanding of their cluster-surface interaction is an important field, particularly with respect to forming cluster-assembled materials. The authors are experts in this field and the paper is well written. While these results deserve to be published, I do not believe *Nature Communications* is the right journal.

First, reactivity of Al_n clusters and in particular the chemical inertness of Al_{13}^- (Fig. 3) is well-known. Similarly, calculations of neutral and charged Al_n and $B@Al_{12}$ clusters (Fig. 5) have been carried out before. What is new here is that the interaction and charge-transfer are shown to depend upon the substrate. As such, this is expected. I recommend a more specialized journal.

(Our reply)

Thank you for this comment. As the reviewer correctly points out, the Al_{13}^- (and $B@Al_{12}^-$) superatom has been historically studied, and its high chemical stability in the gas phase has been reported (Refs. 22 and 23). A number of theoretical studies have also been published related to Al_{13}^- and its related compounds (e.g., $Al_{13}^-K^+$, and Al_{13}^-I ; *J. Chem. Phys.* **124**,144304 (2006) and Ref. 26.) to explain the significant stability. Although the high chemical stability of supported Al_{13}^- is readily expected from analogy with the gas phase experiments, its superatomic nature in the condensed system has yet to be reported or characterized, with the exception of a single report

into the solution-phase synthesis of Al_{13}^- (Ref. 43). The lack of published work in this area can be attributed to the following experimental difficulties:

1. The large-scale generation and size-selective soft-landing (non-destructive) deposition of atomically-controlled nanoclusters on the surface are required;
2. The nanoclusters should be immobilized on the surface without aggregation to allow characterization of their individual electronic and chemical properties;
3. Charged nanoclusters in the gas phase, such as Al_{13}^- , must be preserved in its charged state even on the surface, otherwise it will lose its characteristics.

As described in the main text, we have successfully deposited precisely size-selected nanoclusters on a surface in a non-destructive manner, wherein the abundant deposition amount (2.9×10^{13} superatoms) allows characterization of the electronic structure and chemical robustness through X-ray photoelectron spectroscopy combined with O_2 gas exposure experiments. In addition, we successfully evaluated the deposited nanoclusters based on non-aggregated Al_{13}^- on the substrate surfaces by selecting the appropriate organic molecules for substrate decoration. For the Al_{13}^- superatom, we found a *p*-type HB-HBC is required to preserve its charge state through favorable charge transfer interactions. Moreover, structural information related to Al_{13}^- has been examined by spectroscopic means and through evaluation of the reaction behaviors of the boron-doped B@Al_{12}^- .

Our results obtained for the Al_{13}^- and B@Al_{12}^- species can therefore be considered of important in the context of related nanomaterial fabrication since this methodology is widely applicable to surface decoration/immobilization with superatoms. Therefore, we believe that the present work will be able to bridge the scientific fields of nanocluster superatoms and condensed matter, which will open new routes to the utilization of superatoms as a new functional nanomaterial.

Based on the above points, we believe that our manuscript should be of interest to not only scientists in the field of nanocluster superatoms, but also wider fields related to functional nanomaterial science, and so we would like to cordially ask you to re-consider the rating of our manuscript.

It will be useful to present the ionization potential of $\text{C}_{66}\text{H}_{66}$ and to explain why the charge distribution takes the form $\text{Al}_{13}^+\text{C}_{60}^-$ since the electron affinity of Al_{13} is larger than that of C_{60} .

(Our reply)

Thank you for your kind suggestion. As the reviewer correctly points out, the electron affinity of

Al_{13} ($EA = 3.62$ eV) is larger than that of C_{60} ($EA = 2.683$ eV). However, the stability of the charge transfer complexation is determined by the competitive combination between the electron affinity and the ionization energy; endothermicity from the state of neutrals to the state of the corresponding cation and anion. Indeed, the charge transfer complexation between superatoms and organic molecules is of particular importance, as discussed in Supplementary Note 3 in the original Supplementary Information. Regarding this competitive combination, the ionization energies and electron affinities for the Al_{13} and B@Al_{12} superatoms as well as for the C_{60} and HB-HBC molecules are shown in Supplementary Table S2, wherein both experimental and theoretical values have been provided. Since the ionization energy of Al_{13} ($E_i = 6.42$ eV) is smaller than that of C_{60} ($E_i = 7.57$ eV), the state of $\text{Al}_{13}^+\text{C}_{60}^-$ is 0.21 eV more stable than that of $\text{Al}_{13}^-\text{C}_{60}^+$, as can be seen in Supplementary Table 3 (the endothermic dissociation limits of $\text{Al}_{13}^+\text{C}_{60}^-$ and $\text{Al}_{13}^-\text{C}_{60}^+$ are 3.74 and 3.95 eV, respectively). The corresponding schemes for the charge transfer complexations are shown in Supplementary Fig. 12, and our evaluation was also found to be suitable to describe the stabilities of other superatom–molecule combinations, including $\text{Al}_{13}/\text{B@Al}_{12}$ on HB-HBC, as tabulated in Table S3. Such charge transfer complexation using *n*-type or *p*-type molecular decoration on the substrate is of particular interest and importance in the context of superatom immobilization. In the original manuscript, the description regarding superatom–molecule complexation had been mentioned only in Supplementary Note 3, and then the Supplementary Fig. 12, Table 2, and Table 3 have now been explicitly mentioned in the revised manuscript.

Original manuscript (Page 10, Line 254)

“see Supplementary Note 3”

Revised manuscript (Page 10, Lines 254–255)

“...(see Supplementary Note 3 and related contents, i.e., Supplementary Fig. 12, Table 2, and Table 3).”

We hope that these responses and revisions are satisfactory, and we would like to cordially ask you to re-consider the rating of our manuscript.

Sincerely yours,
Atsushi Nakajima

REVIEWER COMMENTS

Reviewer #1 (Remarks to the Author):

The authors have made efforts to improve the manuscript significantly, including the balanced references, the added STM experiments showing the assembly of Lu@Si16. Although AI13 is not attained, it is believed that the cluster assembly support their conclusion. I recommend an acceptance of this manuscript on Nature Communications.

Reviewer #3 (Remarks to the Author):

The authors have addressed the comments raised by all the reviewers and I am satisfied that the results meet the standard of Nature Communication. I am happy to recommend its publication.

Reviewer #1 (Remarks to the Author)

Comments:

The authors have made efforts to improve the manuscript significantly, including the balanced references, the added STM experiments showing the assembly of Lu@Si16. Although Al13 is not attained, it is believed that the cluster assembly support their conclusion. I recommend an acceptance of this manuscript on Nature Communications.

(Our Reply)

We thank the reviewer for his/her positive comments and recommendation for publication in Nature Communications.

Sincerely yours,
Atsushi Nakajima

Reviewer #3 (Remarks to the Author)

Comments:

The authors have addressed the comments raised by all the reviewers and I am satisfied that the results meet the standard of Nature Communication. I am happy to recommend its publication.

(Our Reply)

We thank the reviewer for his/her positive comments and recommendation for publication in Nature Communications.

Sincerely yours,
Atsushi Nakajima